# Encapsulation of Asparaginase as a Promising Strategy to Improve In Vivo Drug Performance

**DOI:** 10.3390/pharmaceutics13111965

**Published:** 2021-11-19

**Authors:** Francisca Villanueva-Flores, Andrés Zárate-Romero, Alfredo G. Torres, Alejandro Huerta-Saquero

**Affiliations:** 1Centro de Nanociencias y Nanotecnología, Universidad Nacional Autónoma de México, Km. 107 Carretera Tijuana-Ensenada, Ensenada 22860, Mexico; francisca.villanueva@ibt.unam.mx (F.V.-F.); azarate@cnyn.unam.mx (A.Z.-R.); 2Department of Microbiology and Immunology, University of Texas Medical Branch, Galveston, TX 77550, USA; altorres@utmb.edu

**Keywords:** asparaginase, acute lymphoblastic leukemia, biobetter, quality by design, nanocarrier

## Abstract

Asparaginase (ASNase) is a widely applied chemotherapeutic drug that is used to treat Acute Lymphoblastic Leukemia (ALL); however, immune responses and silent inactivation of the drug often limit its bioavailability. Many strategies have been proposed to overcome these drawbacks, including the development of improved formulations (biobetters), but only two of them are currently on the market. Nano- and micro-encapsulation are some of the most promising and novel approaches to enhance in vivo performance of ASNase, preventing the direct contact of the enzyme with the environment, protecting it from protease degradation, increasing the enzymes catalytic half-life, and in some cases, reducing immunogenicity. This review summarizes the strategies, particularly for ASNase nano- and micro-encapsulation, and their main findings, constraints, and current gaps in the state-of-the-art knowledge. The pros and cons of the use of different nanocarriers are discussed with the idea to ultimately provide safer and more effective treatments for patients with ALL.

## 1. Introduction

Acute lymphoblastic leukemia (ALL) is a malignant proliferation of lymphoid cells that occurs when primary white blood cells of lymphoid origin reproduce, without developing into normal B and T cells in the bone marrow, blood, and extra-medullar sites [1]. The American Cancer Society estimates about 5690 new cases and 1580 deaths from ALL in the United States only during 2021 (https://www.cancer.org/cancer/acute-lymphocytic-leukemia, accessed on 1 September 2021).

Conventional treatment for pediatric ALL consists of induction therapy with steroids, vincristine, and L-asparaginase (ASNase) with or without anthracycline, followed by multi-agent consolidation and re-induction therapy [2]. ASNase (EC 3.5.1.1) is a homotetrameric enzyme that catalyzes asparagine (Asn) hydrolysis to aspartate (Asp) and production of ammonia. ASNase reduces the blood Asn levels, leading to starvation-induced apoptosis in leukemic cells, due to their lower expression of asparagine synthetase (Figure 1) [3,4,5,6].

ASNase has been successfully applied for ALL treatment. The survival rate for children with ALL is about 90%, and in contrast, only 30–40% of adult patients with ALL achieve long-term remission [7]. The highest remission rates are obtained in adolescents and young adults with ALL, when treated with pediatric-based regimens which include ASNase, in comparison to adult protocols, which do not include it [8,9,10,11,12]. In the case of adults with ALL (and some young patients), ASNase treatment has shown hypersensitive reactions, unfavorable immune responses, and undesired side effects [13,14]. ASNase contains 7–8 recognition sites on its surface that result in severe immune sensitive reaction [15]. Further, ASNase treatment is often limited by its short circulatory half-live and undesired side effects, due to ammonia accumulation and cross-reacting glutaminase activity [16].

To overcome these limitations, modified biopharmaceuticals and formulations with enhanced properties are required. This special class of biopharmaceuticals is called “biobetters” [17]. Biobetters are molecules that have been structurally and/or functionally modified and show improved efficacy, safety, stability, an enhanced formulation, or dosage regimen (tolerability), when compared to the original drug. Unlike biosimilars, biobetters are patentable and can be launched onto the market even before the original drug patent expires. Moreover, the time and monetary cost for developing biobetters is considerably lower than the reference drug. Together, these advantages have skyrocketed the biobetters investment and research [18].

Current development of biobetters is based on a “Quality by Design” (QbD) approach. QbD is a concept introduced in 1992 by Dr. Joseph M. Juran and consists in products designed with predefined objectives based on sound science and quality risk management. QbD aims to enhance the safety, effectiveness, and improvement of the manufacturing quality performance by reducing possible problems related to the initial designing strategy of the product [19,20]. Based on the QbD approach, many biobetters have been developed including ASNase derivatives. However, despite ASNase relevance in ALL treatment, since the first approved *E. coli* formulation of ASNase (Elspar^®^) by the Food and Drugs Administration (FDA) in 1978, only two biobetters have been commercially available, the PEGylated ASNase (Oncaspar^®^) which was approved in 1994, and an ASNase from *Erwinia chrysanthemi* (Erwinase^®^), approved in 2011 [21,22]. A scheme of the principal strategies for designing ASNase biobetters is presented in Figure 2 and can be summarized as follows:(a)Screening of ASNases from different microorganisms (plant, fungi, bacteria, archaea).(b)Protein engineering mainly focused on reducing the glutaminase cross-reactivity, enhancement of enzyme stability and reducing immunogenicity.(c)Chemical modifications (i.e., crosslinking, covalent binding).(d)Enzyme confinement (i.e., nano- and micro-encapsulation, immobilization in/on a substrate or scaffold) [16,23,24,25,26,27,28,29].

Nano- and micro- encapsulation are some of the most promising and novel approaches to enhance ASNase in vivo performance, because the “cage” prevents the direct contact of the enzyme with the environment, protects it from protease degradation, increases the enzymes catalytic half-life in vivo [24] and, in some cases, it might reduce immunogenicity. A limiting factor in the use of asparaginases for treating ALL is their bacterial source. Furthermore, the bacterial and eukaryotic glycosylation patterns are completely different from each other, so that glycosylation greatly influences the immunogenicity of proteins. It has recently been shown that hypersensitivity to *E. coli* ASNase may be mediated by anti-ASNase IgG and IgE through immunoglobulin receptors FcγRIII and FcεRI, respectively [30]. This is consistent with the fact that specific IgE antibodies for bacterial antigens are found in the serum of allergic individuals [31]. In this context, the nano- or micro-encapsulation of ASNase can be a useful strategy to mask the recognition sites of ASNase, delaying their recognition by the immune system and reducing hypersensitivity. ASNase encapsulation serves to improve in vivo and in vitro enzymatic performance, which is often neglected in the literature, and it is worth reviewing its benefits and limitations. Hence, the goal of this review is to summarize the strategies applied so far for ASNase encapsulation and their main findings. We expect to highlight current gaps in knowledge, as well as possible strategies to overcome those gaps and to face current ASNase limitations for the safer and more effective ALL treatment.

## 2. Carriers for ASNase Encapsulation

To improve the ASNase stability and to reduce its immune response, encapsulation using many carriers comes with specific advantages and limitations. In general, an “ideal” enzyme carrier should be biocompatible, scalable, non-immunogenic, low-cost, and stable; it should favor the enzyme dynamics, not interfering with the catalytic activity and allowing substrate diffusion across the carrier membrane. 

ASNase has been encapsulated in diverse carriers such as liposomes [32,33,34] biopolymers [35,36,37], silk nanoparticles [38,39], hybrid materials [40], polymerosomes [41], magnetic nanoparticles [42], polyion vesicles [43], red blood cells (RBC) [44], hollow nanospheres [45], and virus-like particles (VLPs) [46] (Figure 3).

Each carrier requires specific methodologies for ASNase encapsulation and each one shows advantages and limitations compared to the free enzyme. Table 1 shows a summary of pros and cons for each carrier. A comprehensive and comparative discussion of these is presented next.

## 3. Liposome-ASNase

Liposomes are spherical structures that have an aqueous core enclosed by one or more phospholipids (glycerophospholipids and sphingomyelins) bilayers or lamellae [47]. Liposomes are one of the first particles used for enzyme entrapment [48]. The wide use of liposomes for research is due to the simplicity and low-cost of the conventional methodologies for their production. Similar preparation methods can be used for all lipid vesicles, regardless of their composition. A general procedure involves preparation of the lipids by hydration, hydration with agitation, and sizing to a homogeneous distribution of vesicles. Once formed, liposomes can be easily separated by size via ultracentrifugation (Figure 4). An excellent critical review has been recently published where the most used methods for liposomes preparation are discussed in detail [49]. Lipid based drug carriers are already approved for nanomedical purposes and in clinical trials. Liposomes are usually functionalized to overcome their low stability, leakiness, and low blood circulation times [47,50]. The underlying principle for liposome formation is based on the hydrophobic and hydrophobic–hydrophilic interactions between lipid–lipid and lipid–water molecules, respectively. An input of energy (i.e., sonication, heating, etc.) is required to form bilayer vesicles and to achieve a thermodynamic equilibrium in the aqueous phase [51]. Phospholipids are amphiphilic molecules with a hydrophobic long hydrocarbon chain and a hydrophilic head. In the presence of water, phospholipids are spontaneously self-assembled. Assembly is determined by the energy balance between edge interaction and bending elastic energies [52].

Liposomes have been extensively applied for enzyme-replacement therapy [53]. Jorge JC. et al. (1994) reported that the synthesis of a derivative palmitoyl-L-asparaginase (palmitoyl-ASNase) that was incorporated in liposomes (>200 nm of diameter) and injected into mice that have been previously inoculated with P1534 lymphoma cells. The authors showed that palmitoyl-ASNase maintained the anti-tumoral activity in vivo, increased its blood half-life (from 2.88 h to more than 23.7 h), and prevented acute toxicity when encapsulated in liposomes [33]. Similar results were obtained by Do TT. et al., (2019) who administered 6 IU/mouse of a PEGylated nanoliposomal ASNase (93 nm of diameter) to tumor-induced mice. The authors found that the anti-tumor activity of liposomal ASNase was significantly greater than the free ASNase, in terms of tumor size (6309.11 ± 414.06 mm^3^ versus 7544.94 ± 284.05 mm^3^, respectively) [54]. However, the in vitro cytotoxic activity of PEGylated nanoliposomal ASNase was less effective against several carcinoma cell lines than the free ASNase [32]. The authors discussed that the reduction of ASNase cytotoxicity, when encapsulated in liposomes, might be due to a slower release rate and slower substrate depletion, allowing cells to adapt and synthesize asparagine by themselves. However, the analysis of the kinetic parameters of the enzyme was not shown. Altogether, it highlights the importance of studying the enzymatic profile, particularly for encapsulated enzymes. The encapsulated enzyme operates under mass transfer limiting conditions. The liposome that wraps the enzyme, influences in its catalytic rate, limiting substrate diffusion [55]. Mass-transfer resistance decreases when flowrates and stirring increased [56,57]. Most of the current literature has focused on enzymatic profiles and characterization during resting conditions. The effect of the flowrate is important in the determination of the Michaelis–Menten constant (*Km*) and the reaction’s maximum velocity (*Vmax*) as was showed by Bartolini et al. (2003), who observed that the enzymatic activity of an immobilized D-glyceraldehyde-3-phosphate dehydrogenase enzyme reaction significantly decreased when flow rate increased [57]. In conclusion, studying the kinetic parameters of encapsulated enzymes is critical to evaluating their effects in vivo. It is highly encouraging to perform enzyme kinetic analyses in a fluid, which mimics the biological environment, instead of under conventional resting conditions.

## 4. Liposome Size Influences Blood-Circulating Time

Particle size is an important parameter that affects the ability of liposomes to be internalized into cells and for the in vivo particle biodistribution. Liposome sizes can typically range from 0.05 to 5 µm. Gaspar M.M. et al. (1996) demonstrated the effect of liposome size in the time for circulation in tumor P1534-induced mice. According to their results, large liposomes (1.25 µm of diameter) showed the lowest circulation time of ASNase enzyme, whereas small liposomes (158–180 nm of diameter) prolonged it [34].

One of the drawbacks of the use of liposomes is the fast elimination from the blood and their capture by the cellular reticuloendothelial system. For a successful ALL treatment, a long circulation time in blood of the liposome-ASNase formulation is required. Based on the ability of liposomes to interact with cells or blood components, non-interactive, sterically stabilized liposomes (long-circulating) liposomes (LCL) are preferred. LCL can be formulated by incorporating hydrophilic long-chain polymers (such as PEG or glycolipids) in the bilayer, which form a coat on the liposome surface, avoiding cell adsorption and reducing opsonization [59,60]. Up to date, systematic studies analyzing the impact of the liposome size in the enzyme’s performance are lacking in the literature, but if performed, they could contribute to a better understanding of the pharmacodynamics which will contribute to the optimal design of liposome-ASNase formulations.

## 5. Biopolymer-ASNase

Biopolymers are probably the most used carriers for enzyme encapsulation. They can be obtained at low cost from agricultural waste or through biotechnology, from bacterial cultures, among others. Biopolymers are non-toxic, biocompatible, biodegradable, and non-antigenic molecules, which make them suitable for biomedical applications. Due to their reactive functional groups, biopolymers can be used to couple enzymes with high loading rates and low diffusion. Biopolymer versatility has made it possible to obtain enzyme-biopolymer complexes in diverse formulations, such as nanoparticles, capsules, hydrogels, or films for several applications. In general, entrapment of enzymes onto biopolymers has shown to improve their stability under different ranges of pH and temperature, as well as to provide resistance to proteases and other denaturing compounds. The most studied biopolymers for enzyme entrapment are chitosan, calcium alginate, cellulose, agarose, PLGA, silk fibroin, collagen, and gelatin, among others [61].

ASNase has been encapsulated in diverse biopolymers. For example, Bahreini E. et al. (2014) encapsulated ASNase in particles (247 nm in diameter) made of chitosan, crosslinked with sodium tripolyphosphate (TPP) by using ionic gelation. According to their results, when encapsulated, the ASNase showed 30% less specific activity but also a significantly increased in vitro half-life of about 23 days in a low ionic strength solution and about 6.4 days in a high ionic strength solution. The enzymatic stability at different pH values and temperatures of the encapsulated ASNase was found to be comparable to the free enzyme [37].

In another study, Baran ET. et al. (2002) demonstrated the importance of the carrier on wall permeability. The authors compared the encapsulated ASNase in Poly(3-hydroxybutyrate-co-3-hydroxyvalerate) (PHBV) non-conjugated nanoparticles and heparin low molecular weight conjugated PHBV nanoparticles. According to their results, conjugated PHBV nanoparticles showed higher circulation times in blood and yielded four times higher enzymatic activity compared to non-conjugated nanoparticles. Encapsulation in PHBV nanoparticles also prevented hypersensitivity to ASNase in mice. These results highlight that heparin is a promising molecule for conjugated ASNase carrier with favorable pharmacodynamics performance beyond the traditional PEG [36].

## 6. What Should Be Considered before Using a Biopolymer-Based Carrier for ASNase Encapsulation?

First, special attention should be given to the polymer composition and its affinity for the enzyme as demonstrated by Gaspar MM. et al. (1998). The authors studied ASNase encapsulation in three PLG copolymers, with a 50/50 co-monomer ratio, which differed in their molecular weight and in their free or esterified carboxyl-end groups content. Their results showed that 25% of the free enzyme was adsorbed onto the nanoparticles made of PLG, with carboxyl-end groups through the enzyme’s amino groups. Moreover, this formulation showed a continuous release of the active enzyme for over 20 days and a protein loading of 5% *w*/*w*. In contrast, only 6% of the enzyme interacted with nanoparticles made of PLG with esterified groups. In this case, no enzymatic activity was detected in the release medium after 14 days of incubation. This study indicates that carrier size, encapsulation efficiency, and enzymatic in vitro release properties (enzymatic activity retention and protein quantification) of the nanoparticles are highly affected by the biopolymer molecular weight [35].

Furthermore, large-scale applications of biopolymers have been limited because of their poor mechanical properties. For example, PHBV lacks mechanical strength, wettability, and porosity [62]. Therefore, it is necessary to consider the polymer’s mechanical properties to guarantee its use in real-life conditions.

Finally, the optimal biopolymer-based carrier should exhibit high physicochemical stability, biocompatibility, high purity, an adequate chemical structure for a high binding enzyme rate and homogeneous molecular weight. Most of the natural polymers are polydisperse and show a wide range of molecular weights [63]. Heterogeneity of biopolymer-based formulations has been often neglected in the scientific literature, but it is a very important parameter to obtain a good quality control of the sample in the large-scale fabrication of encapsulated enzymes.

## 7. Silk-ASNase

ASNase has been conjugated to diverse silk derivatives. Silk sericin microparticles were used by Zhang YQ. et al. (2004) to conjugate ASNase with a 62.5% of enzymatic activity recovered. The Km of the ASNase was 8 times lower than the free enzyme and showed a higher thermostability and resistance to trypsin digestion when conjugated to silk sericin microparticles [15].

In a later work, Zhang YQ. et al. (2011) conjugated ASNase to silk-fibroin, and the enzymatic activity recovered after conjugation was nearly 80%. Moreover, ASNase increased heat and storage stability and resistance to trypsin digestion. Unconjugated ASNase was mostly digested, whereas the silk-fibroin ASNase kept over 60% of its original activity. In addition, conjugated ASNase showed a longer half-life (63 h), compared to free ASNase (33 h). The apparent Michaelis constant of the conjugated ASNase (*Km_app_* = 0.84 × 10^−3^ mol L^−1^) was approximately six times lower than the unconjugated enzyme, which suggests that the affinity of ASNase to ASN increased when conjugated with silk fibroin. To determine the immunogenicity of the silk-fibroin ASNase formulations, a counter-immuno-electrophoresis (CIEP) was performed. The CIEP is a technique commonly used to evaluate the binding of an antibody to its antigen in a polyacrylamide gel electrophoresis. In an electric field, the antibody migrates towards its antigen to form a precipitation line, indicating successful binding. No precipitation line was found between antigen, silk fibroin conjugated ASNase, and hyperimmune serum (antibody) obtained from rabbits. These results demonstrate a reduced immunogenicity of ASNase when conjugated to silk fibroin. No in vivo assays were performed in this study [38].

SDS-PAGE analysis revealed that the silk fibroin used by Zhang YQ. et al. (2011) and the silk sericin used by Zhang YQ. et al. (2004), showed molecular weights from 40 to 120 kDa and from 50 to 200 kDa, respectively. Heterogeneous molecular weights could be a drawback that affects entrapment efficiency, stability, and enzyme kinetic parameters at the large-scale manufacturing. Therefore, further studies are needed to clarify this aspect [15,38].

## 8. Polymerosome-ASNase

Polymerosomes are considered as one of the most interesting supramolecular structures for biomedical applications, including drug delivery and generation of artificial organelles that could be integrated in vitro into HeLa cells and in vivo into zebrafish embryos [64]. Polymerosomes are polymeric hollow vesicles enclosing an aqueous cavity; they can be obtained by self-assembly of amphiphilic copolymers. Polymerosomes are composed of two layers of synthetic polymers and show an increased stability compared to liposomes because the polymer membrane helps maintain the encapsulated molecule within the cavity and allows the active molecule to react in the presence of the substrate or when activated by an external trigger [38].

Amphiphilic copolymers that form the polymerosomes can be self-assembled in aqueous solution into different structures depending on concentration, molecular weight, geometry of the amphiphilic polymer, or ratio of the different copolymers. Polymerosomes are promising carriers that show many advantages: (a) polymerosomes size ranges from tens of nm up to μm, (b) their membrane thickness is about 5–50 nm (approximately ten-fold bigger than liposomes), (c) polymerosomes are more stable and permeable than unconjugated liposomes, (d) they can be designed as stimuli-responsive carriers that release their cargo under specific conditions (pH value, temperature, ionic strength, voltage, among others), (e) polymerosomes increase the enzyme’s stability and reduce its immunogenicity, and (f) polymerosomes production is easily scaled *up* [65,66].

Recently, Bueno CZ. et al. (2020) reported the encapsulation of ASNase into hybrid and asymmetric polymerosomes through electroporation [41]. Temporary pores were created by electroporation for ASNase loading, and a 6.6–10% of loading efficiency was obtained. Low loading efficiencies of 11% and 5% have also been reported by Blackman LD. et al. (2018) and Apolinário AC. et al. (2019), respectively [65,67]. Bueno CZ. et al. (2020) generated polymerosomes composed of a mixture of the co-polymers of poly[(2-methacryloyl) ethylphosphorylcholine]-poly[2-(diisopropylamino) ethyl methacrylate] (PMPC-PDPA) (a biocompatible and pH-sensitive copolymer) and poly(ethylene glycol)-poly-(butylene oxide) (PEO-PBO) (which provides permeability to the global polymerosome) [41]. A scheme of the obtained particles is shown in Figure 5. On one hand, the authors of the study discussed that the presence of PEO-PBO favors the ASNase encapsulation and increased the loading volume, when compared to symmetric polymerosomes only composed of PMPC-PDPA, resulting in an increased of enzymatic activity in the asymmetric polymerosomes. Moreover, a higher number of electric pulses in electroporation (20 pulses versus 10 pulses) resulted in higher loading efficiency but also in lower enzymatic activity, which is opposite to a lower number of pulses, because the number of pulses affects the ASNase activity. Although enzyme’s kinetic parameters were no determined, Bueno CZ. et al. (2020) showed interesting evidence for ASNase encapsulation in polymerosomes and generation of polymerosomes with enhanced loading efficiency and good stability for up to two months, which remain a challenge to overcome. Considering the findings of Gaspar MM. et al. (1998), the enzyme’s loading efficiency in polymerosomes could be increased by generating specific interactions between ASNase and the block co-polymers [35]. This can be achieved by focusing particularly on the permeability of polymerosomes and the consequent enzyme release, as was also shown previously by Rideau E. et al. (2018) [66]. Further research should be conducted to take advantage of the polymerosome’s properties to obtain improved formulations with ASNase.

## 9. Magnetic Nanoparticle-ASNase

Magnetic nanoparticles have also gained great attention due to their superparamagnetic properties, high surface area, easy separation, and fate-direction under an external magnetic field. Additionally, they can be functionalized with either organic or inorganic molecules. These properties make the magnetic nanoparticles suitable for clinical diagnostic (magnetic biosensing/imaging) and therapeutic applications, such as drug and gene delivery and hyperthermia therapy [68]. Removal of ASNase-immobilized magnetic nanoparticles from reaction media is an important advantage for their reusability.

Metal oxides have been mostly applied for magnetic nanoparticle synthesis, especially superparamagnetic iron oxide nanoparticles (SPIONs), such as magnetite (Fe_3_O_4_), maghemite (γ-Fe_2_O_3_) and ferrite (15–50 nm of diameter), which can be obtained by chemical, physical, and biological methods [69].

Magnetic nanoparticles have been used as support materials for the binding of enzymes, including ASNase. Magnetic nanoparticles-ASNase have been applied for reducing acrylamide formation in food manufacturing [70] and for biomedical applications [71].

The simplest case of ASNase immobilization onto magnetic nanoparticles is by a direct enzyme-nanoparticle crosslinking, using a standard cross-linker such as glutaraldehyde [72]. Recent studies have been focused on a combination of the magnetic nanoparticle and a stabilizing interface for ASNase immobilization. For example, Teodor E. et al. (2009) immobilized ASNase in biocompatible hydrogel-coated magnetic nanoparticles which were obtained by applying a layer-by-layer technique using hyaluronic acid and chitosan [71]. The obtained nanoparticles sizes were less than 300 nm and 30 nm in the swelled and dried stage, respectively. These conjugates exhibited an immobilization yield of 43–90% and were non-toxic against Vero cells at concentrations between 2 and 12 ng/cell. Similarly, Orhan H. and Uygun DA. (2020) synthesized magnetic nanoparticles (Fe_3_O_4_; diameters between 20–50 nm) coated with poly(hydroxyethyl methacrylate (HEMA)-glycidyl methacrylate (GMA) (poly(HEMA-GMA)). Polymers were used for ASNase immobilization, and according to their results, thermal, storage, and operational stabilities of the ASNase were increased significantly [69]. Moreover, ASNase showed 74.7% of its initial activity in artificial serum. Unfortunately, in vivo assays were not performed in this study.

In addition, inorganic matrixes have also been applied for ASNase immobilization on Fe_3_O_4_ nanoparticles, such as the commercial Mobil Composition of Matter No. 41 (MCM-41), a mesoporous molecular sieve composed of an amorphous silica wall and 3-mercaptopropyltrimethoxysilane (MPTMS) as stabilizer [73]. Immobilized ASNase retained 63% of its original activity and showed a 1.15-fold increased affinity for the substrate when compared to free ASNase. In addition, immobilized ASNase gained pH and thermal stability, kinetic, reusability and storage stability in comparison with the free enzyme. However, no in vivo assays were performed in this study [74].

The in vivo assays of ASNase immobilized onto magnetic nanoparticles are lacking in the literature. We emphasize that a comprehensive understanding of the magnetic nanoparticles interactions with biological systems is required to guarantee their effectiveness and safety. Concerns about magnetic nanoparticles toxicity are based on recent studies that have demonstrated that some magnetic nanoparticles induced inflammation, ulceration, a decrement of the cell growth rate, a decline in viability, and triggering of neurobehavioral alterations in both cell lines and animal models [75,76]. In the short term, toxicity of iron-based magnetic nanoparticles has been attributed to the formation of reactive oxygen species (ROS), including hydroxyl radicals [77]. However, long-term effects remain understudied. A deep understanding about magnetic nanoparticles toxicity is an important gap to overcome to guarantee their safety as scaffold for ASNase administration in ALL preclinical trials.

## 10. Polyion Complex Vesicles-ASNase

Polyion complex vesicles (PICs) are self-assembled molecules formed by a pair of oppositely charged diblock co-polyelectrolytes. PICs are typically obtained by simply mixing oppositely charged macromolecules. A scheme of polyion self-assembly into PICs triggered by an excess of polycation, which elicits the formation of a neutral core is shown in Figure 6 [78].

PICs have recently emerged as novel nanocontainers for substances of biomedical interest. PICs have been studied as carriers for nucleic acids, probes, drug immobilization, and recently for enzyme encapsulation. Kurinomaru T. and Shiraki K. (2015) were the first groups to encapsulate ASNase into polyion vesicles of PEG derivatives [79]. ASNase (anion) and poly(ethylene glycol)-block-poly(N,Ndimethylaminoethyl methacrylate) (PEG-b-PAMA) (cation) formed a water-soluble PEGylated polyelectrolyte. According to the results, composites were able to maintain the enzyme’s fold and activity, also decreasing trypsin digestion and aggregation. No in vivo assays were performed in this study.

In a later study, Sueyoshi D. et al. (2017) have encapsulated ASNase into PICs of 100 nm of diameter. PICs were obtained by dissolving PEG-Poly(α,β-aspartic acid) (PEG-b-P(Asp) and poly([5-aminopentyl] α,β-asoartamide) (P(Asp-AP) in 10 mM phosphate buffer (pH 7.4). The solution was mixed to obtain an equal charge ratio of -COO- and -NH^3+^ units in the side chains of each ionomer and then vortexed to obtain PICs. According to the results, the Km values of the PICs-ASNase and free ASNase were very similar (194 µM and 186 µM, respectively), which indicates that the affinity of ASNase for the substrate is well preserved after encapsulation. Moreover, PICs-ASNase exhibited prolonged blood circulation time compared to free ASNase when administered intravenously to mice [43].

Enzyme encapsulation into PICs is a new, easy, non-expensive and scalable process to improve ASNase performance in vitro and in vivo. Reports of ASNase encapsulated into PICs are scarce in the literature, and there are still many gaps to fill such as improving safety, pharmacokinetics, and pharmacodynamics. In the case of PEGylated ASNase administration, it has been observed that 13% of the patients developed allergies, and 6–8% of the patients developed antibodies (silent inactivation) with no effect of the PEGylated ASNase. An incomplete ASNase therapy is associated with a relapse in the central nervous system [80]. Considering this, it is worth highlighting that rigorous safety assays of PEG-based PICs-ASNase formulations are needed.

Future research focused on the evaluation of novel molecules to form polyion vesicles, beyond PEG derivatives, should be encouraged. Enzyme incorporation into polymerized ionic liquids has gained attention, which are usually prepared by polymerizing ionic liquids monomers. These formulations combined the advantages of ionic liquids (ionic conductivity, thermal, and chemical stability, and tunable solution properties) and the properties of polymers (thermal and chemical stability) [81]. This alternative approach has been poorly explored for enzyme entrapment and could be an interesting alternative for ASNase encapsulation. Further research conducted to explore alternatives for non-covalent ASNase nanoencapsulation could provide valuable information that helps preserve the enzymatic activity, native folding, and freedom of movement inside the cage, which is required to perform catalysis.

## 11. Hollow Nanospheres-ASNase

Bottom-up technologies for generating novel macromolecules have skyrocketed in recent years. Hollow nanospheres are a special class of synthetic nanoparticles with an internal cavity. Hollow nanospheres can be composed by a unique wall or by complex structures with multi-shelled structures and hierarchical architectures to provide specific physicochemical characteristics. Hollow nanospheres have played a notable role in pollutants removal, sensors development, and recently in enzyme encapsulation [83]. Zhu M. et al. (2021) have recently published an excellent review about different approaches for hollow nanospheres synthesis based on hard-, soft-, and self-template and template free approaches [53]. Some advantages and disadvantages of each approach are summarized in Figure 7.

Ha W. et al. (2010) have encapsulated ASNase into soft template-based semi-permeable hollow nanospheres of alginate-graft-poly-(ethylene glycol) (Alg-g-PEG) and α-cyclodextrin (α-CD) with a diameter of 200 nm. The ASNase encapsulation was performed by Alg-g-PEG and ASNase, adding dropwise to the α-CD solution and mixing. Mild conditions for encapsulated ASNase allowed 85.5 ± 5.5% of the free enzyme activity; one of the best activities recovered for ASNase encapsulation in different carriers. Initial rate of the encapsulated ASNase showed a slight decrement in activity when compared to the free enzyme, probably attributed to substrate diffusion interferences across hollow nanospheres pores. Encapsulated ASNase showed significantly higher stability in an acidic environment when compared to the free enzyme. Surprisingly, the *Km* value of encapsulated ASNase was 1.54-fold lower than the *Km* value of the free enzyme, which indicates that the affinity between ASNase and Asn increased when it is encapsulated [45]. No in vivo assays were performed in this study.

In a later study, ASNase was encapsulated in self-assembled hyaluronic acid-graft-poly(ethylene glycol)/hydroxypropyl-beta-cyclodextrin hollow nanospheres, with a diameter of 367.43 ± 2.72 nm. These formulations were evaluated in rats. Results demonstrated that ASNase extended its biological half-life when encapsulated [84]. Evidence of ASNase encapsulation into hollow nanospheres is scarce in the literature, and the in vivo pharmacokinetic, pharmacodynamics, safety, cell uptake, and effectiveness, among others, have not been well established yet for this class of formulations. Therefore, further research is needed to explore the use of other materials based on hard-, soft-, and self-template and template free approaches for hollow nanospheres synthesis.

## 12. Virus-Like Particles-ASNase

Virus-like particles (VLPs) are multiprotein nanostructures that mimic the organization and conformation of viruses but lack the viral genome, and therefore, they cannot replicate in cells. VLPs can be chemically and genetically modified to display antigens in their surface and be used as vaccine carriers. Moreover, VLPs can be loaded with different cargos, such as drugs, genetic material, and more recently, enzymes, so they can be applied as nanocarriers, nanocontainers, or nanoreactors [85,86].

To our knowledge, Díaz-Barriga C. et al. (2021) have reported the only study where ASNase has been encapsulated into VLPs. In this study, ASNase encapsulation was genetically directed in bacteriophage P22-based VLPs. The encapsulation strategy consisted in a double transformation of *E. coli* (BL21 DE3/pLysS) with a vector encoding for the ASNase fused to the scaffold protein (ASNase-SP) and a vector encoding for P22-coat protein (CP). Differential overexpression was performed, first inducing expression of the ASNase-sp, followed by CP expression, which elicit in vivo self-assembly of the ASNase-P22 nanoreactors (Figure 8). The *Km* value of ASNase was 15-fold higher when encapsulated, which indicates that the affinity for the substrate decreased. This observation can be attributed to the limited substrate diffusion across the VLP. ASNase showed an improved resistance to aggregation when encapsulated in P22-based VLPs. Encapsulated ASNase was stable for up to 24 h at 37 ºC, independent of the presence of 10% human blood serum (HBS). ASNase-VLPs were cytotoxic against the leukemic cell line MOLT-4 in a concentration dependent manner [46]. Nonetheless, this is an innovative proof of concept to encapsulate ASNase into VLPs; P22-based VLPs are immunogenic, and therefore, the problem of the intrinsic ASNase immunogenicity is still unsolved. A promising alternative is the use of plant derived VLPs, such as the Brome Mosaic Virus (BMV) for ASNase encapsulation (unpublished data). Interestingly, Nuñez-Rivera A. et al. (2020) have demonstrated that BMV-based VLPs did not activate macrophages in vitro, which suggests that BMV-based VLPs might be less immunogenic, and potentially, it can be applied as non-immunogenic nanocarrier [87]. Further research is being conducted to evaluate this hypothesis.

## 13. Red Blood Cell-ASNase

Red blood cells (RBCs) have been proposed as biocompatible and circulating microbioreactors for therapeutic enzymes since 1973 [88]. Now, engineered RBCs have reached clinical development with some products already in phase III (Rossi L. et al., 2020). RBC show many advantages compared to other nanocarriers such as an in vivo circulating half-life of 19–29 days (compared to 26–30 h of native ASNase from *E. coli*), capacity to avoid plasma clearance due to the action of proteases, and anti-enzyme antibodies formation and renal clearance. RBCs are biodegradable, can circulate throughout the whole body, and lack toxic subproducts and immunogenicity [89,90]. The main strategies that have been used to load therapeutic enzymes inside RBC are listed below, and a general scheme is shown in Figure 9. Enzymes are loaded into RBCs through membrane pores induced by electroporation or osmosis. Then, the membrane is resealed, and the substrate diffuses into the blood circulating RBC, where it is finally degraded.

Enzymes are targeted to the monocyte-macrophage system of the spleen, liver, and bone marrow. Senescent RBCs loaded with enzymes are removed from circulation by macrophages; then, the enzyme is released, and lysosomal macromolecules are degraded.

Therapeutic enzymes are coupled to the RBC membrane where pathological plasma metabolites are catalyzed. Therapeutic enzymes are produced through the transfection of CD34+ hematopoietic precursors cells with lentiviral vectors. Cells are expanded and differentiated, resulting in mature reticulocytes expressing the recombinant enzyme [90].

ASNase encapsulated in RBCs is currently under clinical trials which have shown promising results for ALL treatment and pancreatic cancer [91,92]. It has been observed that a single dose of RBC-ASNase of 150 IU/kg induced depletion in plasmatic Asn for 18.6 days, such as 8 injections of 10,000 IU/m^2^ of free native ASNase from *E. coli*; and 100 IU/kg showed the best safety and efficacy profile for elderly patients. However, median overall survival was 15.8 and 9.7 months in the 100 and 150 IU/kg cohorts, respectively. Decreased incidence and severity of the allergic reactions and coagulation disorders were observed with RBC-ASNase [91,93].

RBCs are much larger (7–8 µm of diameter) than most of the nanocarriers previously described (100–300 nm of diameter) that are currently used for ASNase encapsulation. It is possible that RBCs’ larger size provides the necessary degree of freedom of the ASNase to perform catalysis, which could explain why enzymes encapsulated in RBCs maintain a higher percentage of the initial activity than those encapsulated in other nanovehicles. If all clinical trials are successful in the following stages, it is very likely that RBC-ASNase are the next ASNase biobetter on the market.

## 14. Conclusions

Biobetters development is an attractive growing market for pharmaceutical investment. ASNase biobetters are specially underexploited and represent an excellent alternative to evaluate. Many gaps remain to be filled such as finding an adequate biocompatible carrier for ASNase encapsulation that allows substrate diffusion, provides the required space to perform the enzymatic dynamic, reduces immunogenic effects, and shows a scalable and low-cost manufacturing. A QbD approach will be useful considering all limitations of the current nanocarriers discussed in this review, such as the intrinsic heterogeneity of natural materials for nano- and micro-encapsulation. On one hand, it is important to consider fundamental questions such as whether encapsulation produces any undesirable effect on the enzymatic activity, both affinity and catalytic efficiency. Additionally, it is important to note that enzymatic assays of an encapsulated enzyme under resting conditions are poor representatives of the real-life conditions. We encourage the investigators to perform these analyses under dynamic conditions to mimic blood stream conditions, where ASNase should be active.

On the other hand, we emphasize here the importance of conducting further research with in vivo assays, to guarantee the formulation’s safety and clinical effectiveness in the initial stages of the biobetter development. Especially, when considering that the enzyme cargo can be released from the carrier to the media, eliciting immunological response and, perhaps, toxicity. This concerns have been aspects that are often neglected in the literature and remain a gap to fill in future research.

## Figures and Tables

**Figure 1 pharmaceutics-13-01965-f001:**
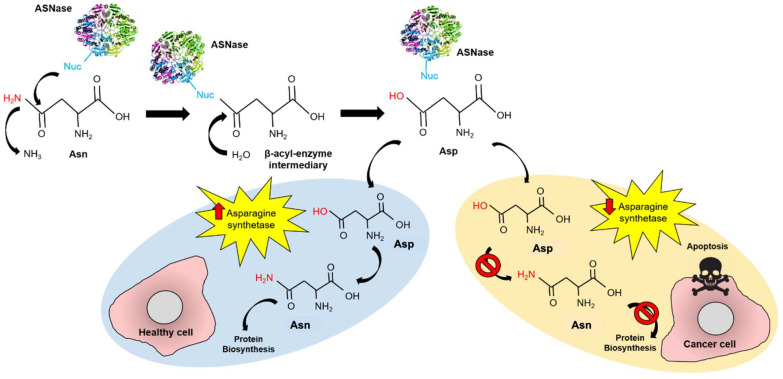
Mechanism of action of ASNase in ALL treatment. ASNase performs a nucleophilic attack to Asn forming the β-acyl-enzyme intermediary, which is rapidly transformed to Asp. In healthy cells, Asp is transformed into Asn by the asparagine synthetase, while protein biosynthesis continues. In cancer cells, asparagine synthetase is under-expressed and the Asn cell-source becomes dependent on circulating Asn uptake. In this way, providing external ASNase into the bloodstream induces apoptosis in cancer cells but not in healthy cells.

**Figure 2 pharmaceutics-13-01965-f002:**
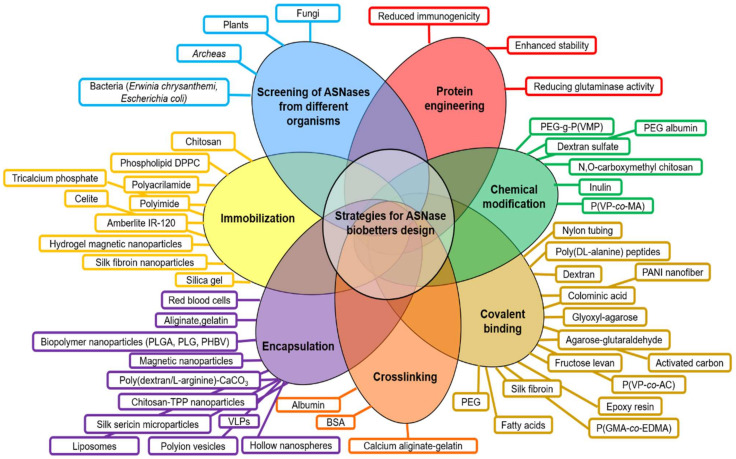
Current strategies for the design of ASNase biobetters. Principal efforts to enhance ASNase clinical performance have been focused on the screening of novel ASNases from different organisms, followed by protein engineering, chemical modifications, covalent binding, immobilization and nano- and micro- encapsulation. Abbreviations: PEG: Poly(ethylene glycol); PEG-g-P(VMP): polyethylene glycol grafted vinylpyrrolidone-maleic anhydride copolymers; P(VP-co-MA): poly(N-vinylpyrrolidone-co-maleic anhydride); PANI: polyaniline; P(VP-co-AC): poly(N-vinylpyrrolidone-co-acrolein); P(GMA-co-EDMA): poly(glycidyl methacrylate-co-ethylene dimethacrylate); BSA: bovine serum albumin; VLPs: virus-like particles; TPP: tripolyphosphate; DPPC: dipalmitoylphosphatidylcholine; PEG, polyethylene glycol; PLGA: poly-(D,L-lactide-co-glycolide); PLG: poly(lactide-co-glycolide); PHBV, poly(3-hydroxybutyrate-co-3-hydroxyvalerate); PVDMA, poly(2-vinyl-4,4-dimethylazlactone); PMMA: poly(methyl methacrylate); pHEMA: poly(2-hydroxyethyl methacrylate); S: starch.

**Figure 3 pharmaceutics-13-01965-f003:**
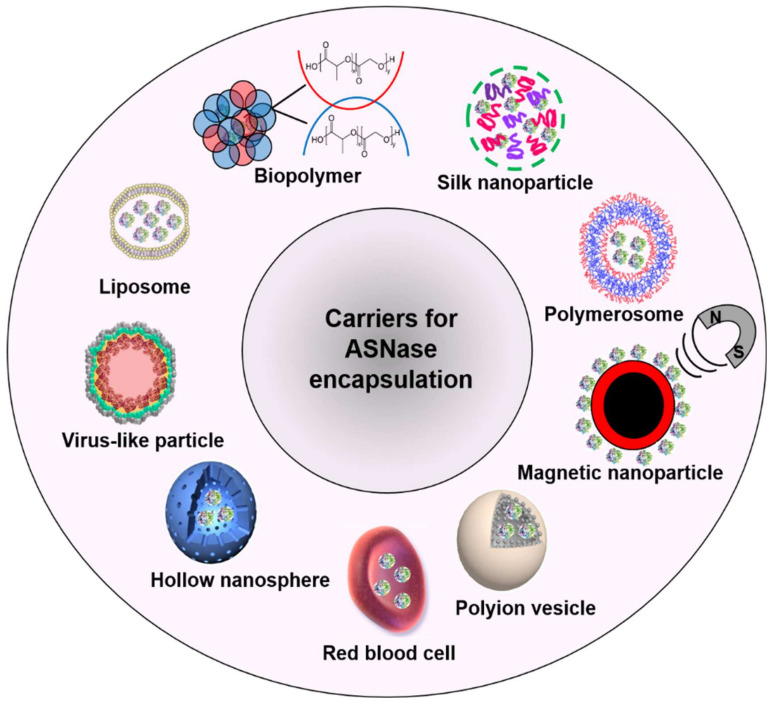
Applied carriers for ASNase encapsulation.

**Figure 4 pharmaceutics-13-01965-f004:**
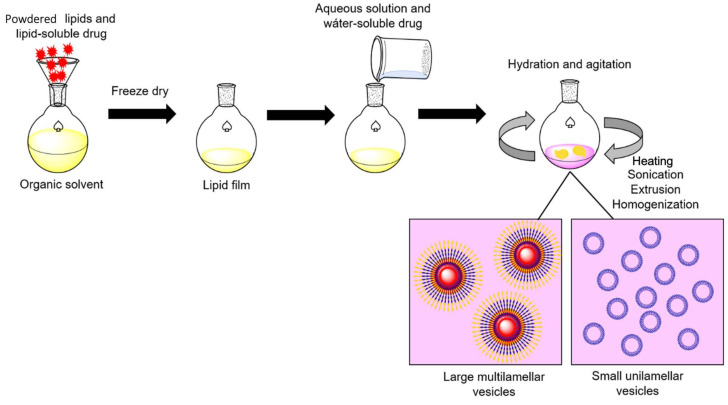
General procedure for enzyme-liposome preparation. Powdered lipids and the lipid-soluble drug are dissolved in an organic solvent to obtain a lipid film. Then, the lipid film is rehydrated in a saline buffer containing the water-soluble drug. After overnight incubation, sonication, heating, extrusion, and homogenization are performed to generate lipid-based vesicles, these can be separated according to the desired size by ultracentrifugation. Adapted from [58], Corrèa ACNTF et al.; 2019.

**Figure 5 pharmaceutics-13-01965-f005:**
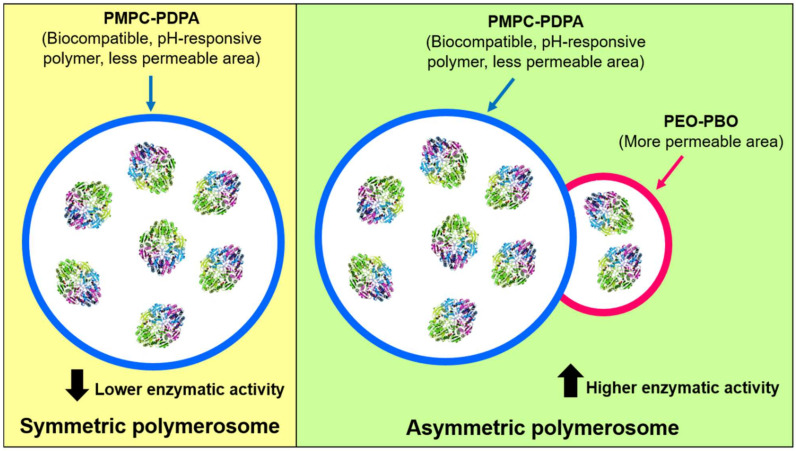
Symmetric and asymmetric polymerosomes composed of PMPC-PDPA polymer and PMPC-PDPA conjugated to PEO-PBO polymer, respectively. The presence of a more permeable zone was provided by the PEO-PBO, increasing ASNase activity of the complete nanobioreactor. Adapted from [41], Bueno CZ. et al.; 2020.

**Figure 6 pharmaceutics-13-01965-f006:**
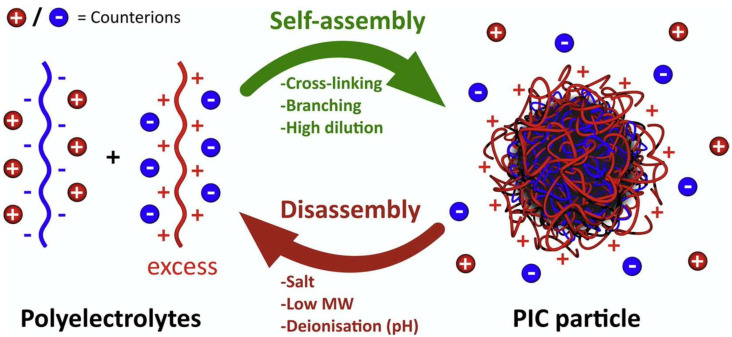
Schematic representation of polyion self-assembly into PICs and subsequent counterion release (red and blue charged spheres). In this example, an excess of polycation triggers the formation of a neutral core. Adapted from [82], Insua I. et al.; 2016.

**Figure 7 pharmaceutics-13-01965-f007:**
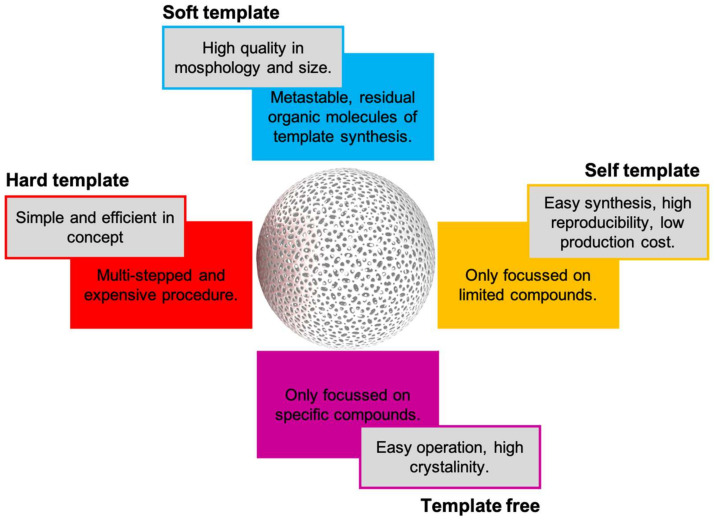
Pros and cons of current synthetic approaches for hollow nanospheres synthesis.

**Figure 8 pharmaceutics-13-01965-f008:**
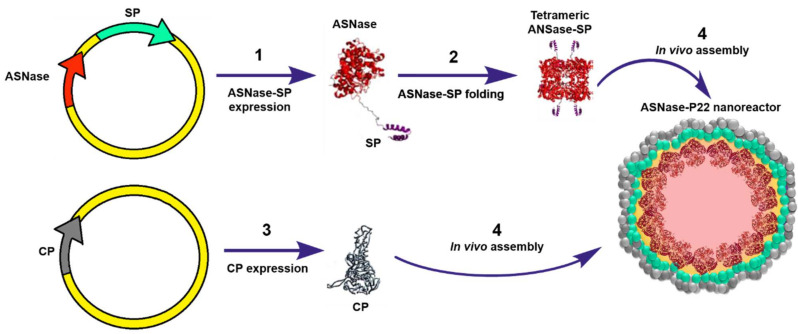
Genetically directed ASNase encapsulation in P22-based VLPs for producing ASNase-P22 nanoreactors. Adapted from [46], Díaz-Barriga C. et al.; 2021.

**Figure 9 pharmaceutics-13-01965-f009:**
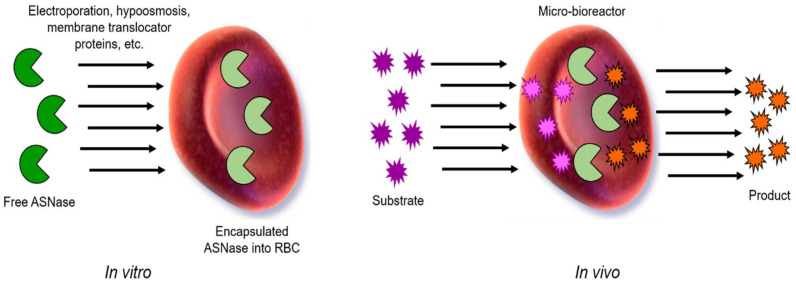
General scheme of encapsulated ASNase into RBCs. Free ASNase can be loaded in vitro into RBCs by electroporation, hypo-osmosis, or by a membrane translocator protein, among others. Then, in vivo, ASNase-RBCs acts as circulating and permeable microbioreactor where the substrate (Asn) enters to the RBCs containing the ASNase, and the product is then released by diffusion into the media. Adapted from [94], Kwon, YM.; 2009.

**Table 1 pharmaceutics-13-01965-t001:** Pros and cons of current carriers used for ASNase encapsulation.

Carriers for Asnase Encapsulation	Advantages	Disadvantages
LIPOSOME	Biodegradable. Approved for clinical trials. Easy synthesis. Low-cost production. Widely studied.	Low stability, leakiness, and low blood circulation times when are not functionalized. Fast elimination from the blood.
BIOPOLYMER	Biodegradable. Can be obtained at low cost from agricultural waste or through biotechnology, from bacterial cultures. Non-antigenic molecules.	Poor mechanical properties. Heterogeneous molecular weights could be a drawback that affects entrapment efficiency, stability, and enzyme kinetic parameters at the large-scale manufacturing.
SILK NANOPARTICLE	Biodegradable. Non-antigenic molecules. Low-cost production.	Heterogeneous molecular weights.
POLYMEROSOME	Can range from nm up to μm. Their membrane thickness is bigger than liposomes. More stable and permeable than unconjugated liposomes. They can be designed as stimuli-responsive carriers. Polymerosomes production is easily scaled up.	Non-biodegradable. Lack of enough information about in vivo cytotoxicity.
MAGNETIC NANOPARTICLE	Easy separation, and fate-direction under an external magnetic field.	Non-biodegradable. Some magnetic nanoparticles induced inflammation, ulceration, and a decrement of the cell growth rate, a decline in viability and triggering of neurobehavioral alterations in both cell lines and animal models.
POLYION VESICLE	Easy, non-expensive and scalable production. These formulations combined the advantages of ionic liquids (ionic conductivity, thermal, and chemical stability, tunable solution properties), and the properties of polymers.	Not always biodegradable. Poorly studied.
HOLLOW NANOSPHERE	Non immunogenic. Scalable process.	Poorly studied.
VIRUS-LIKE PARTICLE	Biodegradable. Scalable and low-cost production. High stability.	Immunogenic when are not functionalized. Poorly studied.
RED BLOOD CELL	Biodegradable, approved for clinical trials. Long blood circulating times. Non immunogenic. Widely studied.	Low rate of red blood cell survival after enzyme internalization. Its production requires permission for human samples managing.

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
