# Peer review of "Encapsulation of Asparaginase as a Promising Strategy to Improve In Vivo Drug Performance"

_pharmaceutics, 2021, doi:10.3390/pharmaceutics13111965_

Round 1
Reviewer 1 Report
Overall, the manuscript is interesting. I like the topic, and the research is thorough. My major issue is that despite it being emphasized in the title and abstract, the QbD approach is barely discussed. It is given a brief mention and definition in the introduction, and mentioned once in the conclusions, but it is not discussed throughout the manuscript. Either the manuscript needs to be significantly modified to include discussion on QbD, or references to it need to be removed entirely from the manuscript. If QbD is kept in the manuscript, it needs to be more thoroughly defined in the introduction. The processes that are part of QbD need to be defined and explained (defining a QTPP, CQAs, risk analysis, CPPs, etc.). What would a QTPP for an ASNase biobetter look like?
In the body of the manuscript, in depth discussions need to be added regarding how QbD can be utilized to develop better ASNase biobetters. One major item highlighted throughout the manuscript is that toxicity is a concern for several biobetters (magnetic nanoparticles come to mind in particular). How could QbD be utilized to address this toxicity risk and develop a product that is non-toxic? Other major issues with nanotechnology in general are cost and scalability. How could QbD be used to develop manufacturing processes that were inexpensive and scalable? For each type of biobetter nanocarrier discussed (liposome, polymerosome, hollow nanosphere, etc.) what are the most important parameters that impact final product quality (i.e. what are the CPPs)? What are the CQAs? Has anyone reported a QbD approach for the development off an ASNase biobetter? Has anyone reported a QbD approach for any type of ALL treatment? If so, these sources need to be cited and discussed in the manuscript.
Content Comments:
- Around line 44, I am curious to know why ASNase is tolerated better in children than adults. Why do children and adults behave differently to treatment?
- Around line 98, I am confused as to how encapsulation of ASNase to “prevent direct contact of the enzyme with the environment” is beneficial. If I understood correctly, ASNase reduces ASN levels in the blood, not in a specific target cell, so wouldn’t the ASNase need to be recognized by blood circulating ASN? Other benefits of encapsulation (reduced immunogenicity, increased half-life, improved stability, etc.) make sense to me, but the “trojan horse” concept is confusing me in this case because I don’t believe there is a target cell to deliver to. I think it would be helpful to clarify early on in the text specifically where the target is for ASNase, if there is a specific target. Is a specific cell type (cancer cell) being targeted or is the drug meant to react with ASN in the blood?
- In figure 4, it would useful to mention that particles can be separated by size via centrifugation in the figure itself, and not just in the figure legend. It would also help if heating was listed as an option (along with sonication, extrusion, and homogenization) in the figure, since this is mentioned as an approach for forming the liposomes in the text (line 137).
- The explanation for liposome formation is not described in enough detail (lines 134-138). It would also be useful to describe the most important parameters that need to be optimized in order to form a stable liposome. Are any preparation methods preferable over the others (for example, is heat more commonly used than sonication), and if so, why?
- Lines 152-169, I think this is a great discussion. I like the emphasis on the importance of better studying enzyme kinetics and adjusting the environment to mimic in vivo conditions. The text is a bit difficult to follow and could use some editing, but the content is good. Also please include the authors cited as reference 53 in line 163.
- Since polymerosomes are formed via self-assembly, would this make them simpler to scale-up compared to liposomes? If so, this could be included as a benefit of polymerosomes around line 297 as a point (f).
- Lines 291 – 293; why is it beneficial that polymerosomes can range up to microns in size? Earlier it was stated that sizes of 150-180nm minimizes recognition by the reticuloendothelial system, so why would a large particle size be beneficial? What is the benefit of having a thick membrane (point b)? Does this improve drug retention?
- Regarding the asymmetric polymerosomes highlighted in figure 5, were any studies performed to evaluate the shelf life and/or stability of these in comparison to symmetric polymerosomes? Does the asymmetric shape impact stability in any way?
Grammatical/Organizational Comments:
- It may be my computer, but it would be beneficial if the figures were higher resolution.
- The sentence beginning on line 185 could be part of the previous paragraph. It does not need to be its own paragraph.
- Line 210, define TPP
- Line 215 should read “importance of the carrier wall permeability”
- Line 462 – encapsulation is misspelled
- Lines 496 – 506; I think these should be listed as bullet points instead of individual one-sentence paragraphs
Author Response
Manuscript ID: pharmaceutics-1418781
Encapsulation of Asparaginase as a Promising Strategy to Improve in vivo Drug Performance: A Quality-by-Design Approach
Francisca Villanueva-Flores, Andrés Zárate-Romero, Alfredo G. Torres, Alejandro Huerta-Saquero *
We want to acknowledge the time and effort the reviewers dedicated to our work. Their suggestions, comments and corrections substantially improved our review. We consider that we have addressed all their suggestions and, we have enhanced the review adding a table and 5 more references.
Reviewer 1.
Comments and Suggestions for Authors
Overall, the manuscript is interesting. I like the topic, and the research is thorough. My major issue is that despite it being emphasized in the title and abstract, the QbD approach is barely discussed. It is given a brief mention and definition in the introduction, and mentioned once in the conclusions, but it is not discussed throughout the manuscript. Either the manuscript needs to be significantly modified to include discussion on QbD, or references to it need to be removed entirely from the manuscript. If QbD is kept in the manuscript, it needs to be more thoroughly defined in the introduction. The processes that are part of QbD need to be defined and explained (defining a QTPP, CQAs, risk analysis, CPPs, etc.). What would a QTPP for an ASNase biobetter look like?
In the body of the manuscript, in depth discussions need to be added regarding how QbD can be utilized to develop better ASNase biobetters. One major item highlighted throughout the manuscript is that toxicity is a concern for several biobetters (magnetic nanoparticles come to mind in particular). How could QbD be utilized to address this toxicity risk and develop a product that is non-toxic? Other major issues with nanotechnology in general are cost and scalability. How could QbD be used to develop manufacturing processes that were inexpensive and scalable? For each type of biobetter nanocarrier discussed (liposome, polymerosome, hollow nanosphere, etc.) what are the most important parameters that impact final product quality (i.e. what are the CPPs)? What are the CQAs? Has anyone reported a QbD approach for the development off an ASNase biobetter? Has anyone reported a QbD approach for any type of ALL treatment? If so, these sources need to be cited and discussed in the manuscript.
Authors’ response:
Thank you very much for your accurate observation. We agreed with it and in order to maintain the focus of the review on the state of the art of different encapsulation techniques using ASNase as the load to achieve enhanced biobetter formulations, we have decided to eliminate the concept of QbD from the main title and to reduce its importance in the manuscript.
Original: “Encapsulation of Asparaginase as a Promising Strategy to Improve in vivo Drug Performance. A Quality by Design Approach.”
Edited: “Encapsulation of Asparaginase as a Promising Strategy to Improve in vivo Drug Performance.”
Content Comments:
- Around line 44, I am curious to know why ASNase is tolerated better in children than adults. Why do children and adults behave differently to treatment?
Authors’ response:
It is certainly an interesting topic. Some authors have proposed some reasons to explain the different susceptibility to ALL when comparing children with adults. Here are a couple of hypotheses:
- ASNase is particularly important in improving the treatment of T-cell childhood acute lymphoblastic leukemia (ALL). However, in adults, only 25% of ALL patients have the T-cell ALL subtype (1).
- There are unspecified chromosomal abnormalities that increase with age. These subtle genetic differences may be related to differential susceptibility to the disease and may explain the increased relapse rate in adults (2).
- Burke PW, Hoelzer D, Park JH, Schmiegelow K, Douer D. Managing toxicities with asparaginase-based therapies in adult ALL: summary of an ESMO Open-Cancer Horizons roundtable discussion. ESMO Open. 2020. 5(5):e000858. doi: 10.1136/esmoopen-2020-000858
- Quist-Paulsen P, Toft N, Heyman M, Abrahamsson J, Griškevičius L, Hallböök H, Jónsson ÓG, Palk K, Vaitkeviciene G, Vettenranta K, Åsberg A, Frandsen TL, Opdahl S, Marquart HV, Siitonen S, Osnes LT, Hultdin M, Overgaard UM, Wartiovaara-Kautto U, Schmiegelow K. T-cell acute lymphoblastic leukemia in patients 1-45 years treated with the pediatric NOPHO ALL2008 protocol. Leukemia. 2020. 34(2):347-357. doi: 10.1038/s41375-019-0598-2
_____________________________________________________________________________________________
- Around line 98, I am confused as to how encapsulation of ASNase to “prevent direct contact of the enzyme with the environment” is beneficial. If I understood correctly, ASNase reduces ASN levels in the blood, not in a specific target cell, so wouldn’t the ASNase need to be recognized by blood circulating ASN? Other benefits of encapsulation (reduced immunogenicity, increased half-life, improved stability, etc.) make sense to me, but the “trojan horse” concept is confusing me in this case because I don’t believe there is a target cell to deliver to. I think it would be helpful to clarify early on in the text specifically where the target is for ASNase, if there is a specific target. Is a specific cell type (cancer cell) being targeted or is the drug meant to react with ASN in the blood?
Authors’ response:
As you mentioned, ASNase does not require a specific molecular target. However, it can be recognized by the immune system.
We rewrite this section as follows:
“Nano- and micro- encapsulation is one of the most promising and novel approaches to enhance ASNase in vivo performance, because the “cage” prevents the direct contact of the enzyme with the environment, protects it from protease degradation, increases the enzymes catalytic half-life in vivo [24] and, in some cases, it might reduce immunogenicity. A limiting factor in the use of asparaginases for treating ALL is their bacterial source. Furthermore, the bacterial and eukaryotic glycosylation patterns are completely different from each other, so that glycosylation greatly influences the immunogenicity of proteins. It has recently been shown that hypersensitivity to E. coli ASNase may be mediated by anti-ASNase IgG and IgE through immunoglobulin receptors FcRIII and FcRI, respectively [30]. This is consistent with the fact that specific IgE antibodies for bacterial antigens are found in the serum of allergic individuals [31]. In this context, the nano or microencapsulation of ASNase can be a useful strategy to mask the recognition sites of ASNase, delaying their recognition by the immune system and reducing hypersensitivity. ASNase encapsulation serves to improve in vivo and in vitro enzymatic performance, which is often neglected in the literature, and it is worth reviewing its benefits and limitations.
_____________________________________________________________________________________________
- In figure 4, it would useful to mention that particles can be separated by size via centrifugation in the figure itself, and not just in the figure legend. It would also help if heating was listed as an option (along with sonication, extrusion, and homogenization) in the figure, since this is mentioned as an approach for forming the liposomes in the text (line 137).
Authors’ response:
Thank you. Your suggestions have been incorporated in the figure.
_____________________________________________________________________________________________
- The explanation for liposome formation is not described in enough detail (lines 134-138). It would also be useful to describe the most important parameters that need to be optimized in order to form a stable liposome. Are any preparation methods preferable over the others (for example, is heat more commonly used than sonication), and if so, why?
Authors’ response:
We added information and this section was re-written as follows:
“Liposomes are spherical structures that have an aqueous core enclosed by one or more phospholipids (glycerophospholipids and sphingomyelins) bilayers or lamellae [47]. Liposomes are one of the first particles used for enzyme entrapment [48]. The wide use of liposomes for research is due to the simplicity and low-cost of the conventional methodologies for their production. Similar preparation methods can be used for all lipid vesicles, regardless of their composition. A general procedure involves preparation of the lipids by hydration, hydration with agitation, and sizing to a homogeneous distribution of vesicles. Once formed, liposomes can be easily separated by size via ultracentrifugation (Fig. 4). An excellent critical review has been recently published where the most used methods for liposomes preparation are discussed in detail [49]. Lipid based drug carriers are already approved for nanomedical purposes and in clinical trials. Liposomes are usually functionalized to overcome their low stability, leakiness, and low blood circulation times [47, 50]. The underlying principle for the liposome formation is based on the hydrophobic, and hydrophobic-hydrophilic interactions between lipid-lipid and lipid-water molecules, respectively. An input of energy (i. e.; sonication, heating, etc.) is required to form bilayer vesicles and to achieve a thermodynamic equilibrium in the aqueous phase [51]. Phospholipids are amphiphilic molecules with a hydrophobic long hydrocarbon chain and a hydrophilic head. In the presence of water, phospholipids are spontaneously self-assembled. Assembly is determined by the energy balance between the edge interaction and bending elastic energies [52]”.
_____________________________________________________________________________________________
- Lines 152-169, I think this is a great discussion. I like the emphasis on the importance of better studying enzyme kinetics and adjusting the environment to mimic in vivo conditions. The text is a bit difficult to follow and could use some editing, but the content is good. Also please include the authors cited as reference 53 in line 163.
Authors’ response:
Thank you. Your suggestions have been incorporated in the text.
_____________________________________________________________________________________________
- Since polymerosomes are formed via self-assembly, would this make them simpler to scale-up compared to liposomes? If so, this could be included as a benefit of polymerosomes around line 297 as a point (f).
Authors’ response:
We agreed. Your suggestions have been incorporated in the text.
_____________________________________________________________________________________________
- Lines 291 – 293; why is it beneficial that polymerosomes can range up to microns in size? Earlier it was stated that sizes of 150-180nm minimizes recognition by the reticuloendothelial system, so why would a large particle size be beneficial? What is the benefit of having a thick membrane (point b)? Does this improve drug retention?
Authors’ response:
Obtaining larger polymerosomes is considered an advantage from the point of view of the versatility of the material. Larger sizes of polymerosomes make it possible to generate on-demand carriers, without limitation of size, as is the case with other nanocarriers, for biomedical applications.
_____________________________________________________________________________________________
- Regarding the asymmetric polymerosomes highlighted in figure 5, were any studies performed to evaluate the shelf life and/or stability of these in comparison to symmetric polymerosomes? Does the asymmetric shape impact stability in any way?
Authors’ response:
Asymmetric polymerosomes were stable for up two months. This data has been included in the manuscript.
_____________________________________________________________________________________________
Grammatical/Organizational Comments:
- It may be my computer, but it would be beneficial if the figures were higher resolution.
Authors’ response:
Thank you, the resolution of figures has been improved.
_____________________________________________________________________________________________
- The sentence beginning on line 185 could be part of the previous paragraph. It does not need to be its own paragraph.
Authors’ response:
Corrected.
_____________________________________________________________________________________________
- Line 210, define TPP
Authors’ response:
Corrected.
_____________________________________________________________________________________________
- Line 215 should read “importance of the carrier wall permeability”
Authors’ response:
Corrected.
_____________________________________________________________________________________________
- Line 462 – encapsulation is misspelled
Authors’ response:
Corrected.
_____________________________________________________________________________________________
- Lines 496 – 506; I think these should be listed as bullet points instead of individual one-sentence paragraphs
Authors’ response:
Corrected.
_____________________________________________________________________________________________
Reviewer 2 Report
Ms. No.: pharmaceutics-1418781
Title: “Encapsulation of Asparaginase as a Promising Strategy to Improve in vivo Drug Performance: A Quality-by-Design Approach”
This is a very important, interesting, and up to date study, in well-written, logically constructed and readable form. Authors aimed to give a comprehensive overview about the state of the art of different encapsulation techniques using ASNase as the load. The goal of these new nanotechnologies is to achieve biobetter formulations.
The review is based on 91 literature studies, out of the 33 are from the recent four years. The figures are very informative and the merit of the study is the critical evaluation of each approach. Reviewer has only two suggestions.
1) An explaining, schematic figure to the Hallow-nanospheres chapter would be useful.
2) A summarising table about the discussed 8 encapsulation techniques would further help the readers’ understanding.
All in all, it is an excellent manuscript, what I suggest for publication.
Some corrections are needed:
- line 35: ammonia (not ammonium)
- line 210: give the meaning of TPP
- line 397: number 3 has to be in lower-script not in upper-script
Author Response
Reviewer 2.
Manuscript ID: pharmaceutics-1418781
Encapsulation of Asparaginase as a Promising Strategy to Improve in vivo Drug Performance: A Quality-by-Design Approach
Francisca Villanueva-Flores, Andrés Zárate-Romero, Alfredo G. Torres, Alejandro Huerta-Saquero *
We want to acknowledge the time and effort the reviewers dedicated to our work. Their suggestions, comments and corrections substantially improved our review. We consider that we have addressed all their suggestions and, we have enhanced the review adding a table and 5 more references.
Title: “Encapsulation of Asparaginase as a Promising Strategy to Improve in vivo Drug Performance: A Quality-by-Design Approach”
This is a very important, interesting, and up to date study, in well-written, logically constructed and readable form. Authors aimed to give a comprehensive overview about the state of the art of different encapsulation techniques using ASNase as the load. The goal of these new nanotechnologies is to achieve biobetter formulations.
The review is based on 91 literature studies, out of the 33 are from the recent four years. The figures are very informative and the merit of the study is the critical evaluation of each approach. Reviewer has only two suggestions.
- An explaining, schematic figure to the Hallow-nanospheres chapter would be useful.
Authors’ response:
Thank you very much for your comments. A schematic figure summarizing the advantages and disadvantages of current synthetic approaches for hollow nanospheres synthesis are included in the main text.
_____________________________________________________________________________________________
2) A summarising table about the discussed 8 encapsulation techniques would further help the readers’ understanding.
Authors’ response:
Thank you. Your valuable suggestion has been incorporated in the main text.
_____________________________________________________________________________________________
All in all, it is an excellent manuscript, what I suggest for publication.
Some corrections are needed:
- line 35: ammonia (not ammonium)
Authors’ response:
Corrected.
_____________________________________________________________________________________________
- line 210: give the meaning of TPP
Authors’ response:
Corrected.
_____________________________________________________________________________________________
- line 397: number 3 has to be in lower-script not in upper-script
Authors’ response:
Corrected.
_____________________________________________________________________________________________
Reviewer 3 Report
pharmaceutics-1418781
Encapsulation of Asparaginase as a Promising Strategy to Improve in vivo Drug Performance: A Quality-by-Design Approach
In this review, the authors summarized proposed strategies for encapsulation of Aspargianse to overcome the drawbacks and improvement common treatments with this enzyme. The topic is very interesting and suitable for the Journal, I have only a few points to discuss.
The quality of figures have to be improved
There is no necessity of providing the following terms ‘Quality by Design’ and ‘Biobeterrs’ since they are only discussed in the Intro and Conclusion parts. However, it is a subjective assessment of the reviewer
Author Response
Reviewer 3.
Manuscript ID: pharmaceutics-1418781
Encapsulation of Asparaginase as a Promising Strategy to Improve in vivo Drug Performance: A Quality-by-Design Approach
Francisca Villanueva-Flores, Andrés Zárate-Romero, Alfredo G. Torres, Alejandro Huerta-Saquero *
We want to acknowledge the time and effort the reviewers dedicated to our work. Their suggestions, comments and corrections substantially improved our review. We consider that we have addressed all their suggestions and, we have enhanced the review adding a table and 5 more references.
Encapsulation of Asparaginase as a Promising Strategy to Improve in vivo Drug Performance: A Quality-by-Design Approach
In this review, the authors summarized proposed strategies for encapsulation of Aspargianse to overcome the drawbacks and improvement common treatments with this enzyme. The topic is very interesting and suitable for the Journal, I have only a few points to discuss.
The quality of figures have to be improved
Authors’ response:
Corrected.
_____________________________________________________________________________________________
There is no necessity of providing the following terms ‘Quality by Design’ and ‘Biobeterrs’ since they are only discussed in the Intro and Conclusion parts. However, it is a subjective assessment of the reviewer
Authors’ response:
Thank you so much for your accurate observation. We agreed and have decided to eliminate the concept of QbD from the main title.
_____________________________________________________________________________________________
Reviewer 4 Report
This is an interesting and comprehensive review on the nano- and micro-encapsulation strategies for asparaginase delivery. The manuscript is well written. Advantages and disadvantages of several methodologies are discussed. There is a very nice review of existing literature and several representative examples are presented.
I suggest improving quality/resolution of the figures, if possible.
Additionally, the schemes presented in Figures 4 and 6 are too general and more specific examples should be illustrated in them.
Author Response
Reviewer 4.
Manuscript ID: pharmaceutics-1418781
Encapsulation of Asparaginase as a Promising Strategy to Improve in vivo Drug Performance: A Quality-by-Design Approach
Francisca Villanueva-Flores, Andrés Zárate-Romero, Alfredo G. Torres, Alejandro Huerta-Saquero *
We want to acknowledge the time and effort the reviewers dedicated to our work. Their suggestions, comments and corrections substantially improved our review. We consider that we have addressed all their suggestions and, we have enhanced the review adding a table and 5 more references.
This is an interesting and comprehensive review on the nano- and micro-encapsulation strategies for asparaginase delivery. The manuscript is well written. Advantages and disadvantages of several methodologies are discussed. There is a very nice review of existing literature and several representative examples are presented.
I suggest improving quality/resolution of the figures, if possible.
Authors’ response:
Thank you for your comments. Figures have been corrected.
_____________________________________________________________________________________________
Additionally, the schemes presented in Figures 4 and 6 are too general and more specific examples should be illustrated in them.
Authors’ response:
Figures have been corrected.
_____________________________________________________________________________________________